

# Paleomass for R—bracketing body volume of marine vertebrates with 3D models

Ryosuke Motani

Departement of Earth and Planetary Sciences, University of California, Davis, Davis, California, United States of America

## ABSTRACT

Body mass is arguably the most important characteristic of an organism, yet it is often not available in biological samples that have been skeletonized, liquid-preserved, or fossilized. The lack of information is especially problematic for fossil species, for which individuals with body mass information are not available anywhere. Multiple methods are available for estimating the body mass of fossil terrestrial vertebrates but those for their marine counterparts are limited. Paleomass is a software tool for estimating the body mass of marine vertebrates from their orthogonal silhouettes through bracketing. It generates a set of two 3D models from these silhouettes, assuming superelliptical body cross-sections with different exponent values. By setting the exponents appropriately, it is possible to bracket the true volume of the animal between those of the two models. The original version phased out together with the language platform it used. A new version is reported here as an open-source package based on the R scripting language. It inherits the underlying principles of the original version but has been completely rewritten with a new architecture. For example, it first produces 3D mesh models of the animal and then measures their volumes and areas with the VCG library, unlike the original version that did not produce a 3D model but instead computed the volume and area segment by segment using parametric equations. The new version also exports 3D models in polygon meshes, allowing later tests by other software. Other improvements include the use of NACA foil sections for hydrofoils such as flippers, and optional interpolation with local regression. The software has a high accuracy, with the mean absolute errors of 1.33% when the silhouettes of the animals are known.

# INTRODUCTION

Body mass is an essential metric to describe aspects of the biology of individual organisms (*Schmidt-Nielsen, 1984*). Despite the importance, a body mass record is not always available—preserved specimens in museum collections often lack body mass information, and fossil organisms are never found with body mass data. The lack of information is not overly problematic for extant species for which conspecific individuals are available elsewhere, but poses a critical hurdle to biological studies of fossil species. Accordingly, paleontologists have been exploring the possibility of body mass estimation based on what is preserved in fossils.

Corresponding author
Ryosuke Motani,
rmotani@ucdavis.edu

Methods for body mass estimation based on fossils are largely divided into two categories depending on the underlying principle—one may be called the length-based and the other volumetric approaches (*Hurlburt, 1999*; *Smith, 2002*; *Sellers et al., 2012*; *Brassey, 2016*; *Campione & Evans, 2020*). The length-based approach first establishes a correlation between the length(s) of one or more morphological character(s) of the animals in question and their body mass through a linear regression, based on extant samples for which the body mass is known, and then uses the regression equation to estimate the body mass of extinct animals for which the length character(s) are available. Multiple regression with more than one length characters tends to be preferred in clades in which fossils species are nested among abundant extant members, such as mammals (*e.g.*, *Smith, 2002*; *Mendoza, Janis & Palmqvist, 2006*), whereas bivariate regression is almost exclusively used in clades that have long been extinct with only distantly related descendants surviving, *e.g.*, non-avialan dinosaur clades (*Anderson, Hall-Martin & Russell, 1985*; *Campione & Evans, 2012*), probably to avoid overfitting of the model to particular extant clades that would mislead the outcome.

The volumetric approach first estimates the volume of the animal in question and then converts the value to body mass by assuming an average body density. This approach dates back at least to 1905, when the body mass of *Brontosaurus* was estimated by measuring the volume of a cast of a scaled physical model with water displacement and then converting the volume to mass by assuming the freshwater density (*Gregory, 1905*). A similar method was used by *Colbert (1962)* for body mass estimation of a wider range of dinosaurs. As mathematical models became common, a parametric approach to model the body as a collection of cylinders based on a limited number of measurements, called Graphic Double Integration, was developed (*Jerison, 1973*). What may be considered an extension of this approach, where the body is straightened in a parametric space and modeled by many cylindrical disks, was later proposed (*Seebacher, 2001*). With the arrival of 3D computer technology, methods of incorporating complex 3D computer models emerged. Such methods include a partly parametric approach as in Paleomass based on superelliptical cross-sections (*Motani, 2001*), as well as the minimum convex hull method based on completely empirical data from laser scanning of mounted skeletons (*Sellers et al., 2012*).

These methods aim to arrive at the best mean estimate of body mass, except Paleomass which tried to bracket the mass between two values (Fig. 1D bracketed by 1C and 1E). The method was also unique for specifically addressing marine vertebrates, for which a limb-based regression approach is not suitable because they do not support the body mass with the limbs. Despite the uniqueness that would allow cross-checking of other methods, the software is no longer available because its language platform was discontinued. The purpose of the present paper is to report a completely rewritten and open-source version of Paleomass with a new architecture and enhancements over the original version.

## MATERIALS AND METHODS

### Platform

The new Paleomass was written in the R scripting language and run on the R platform (*R Core Team, 2020*). Apart from the default R packages, it relies on the following packages
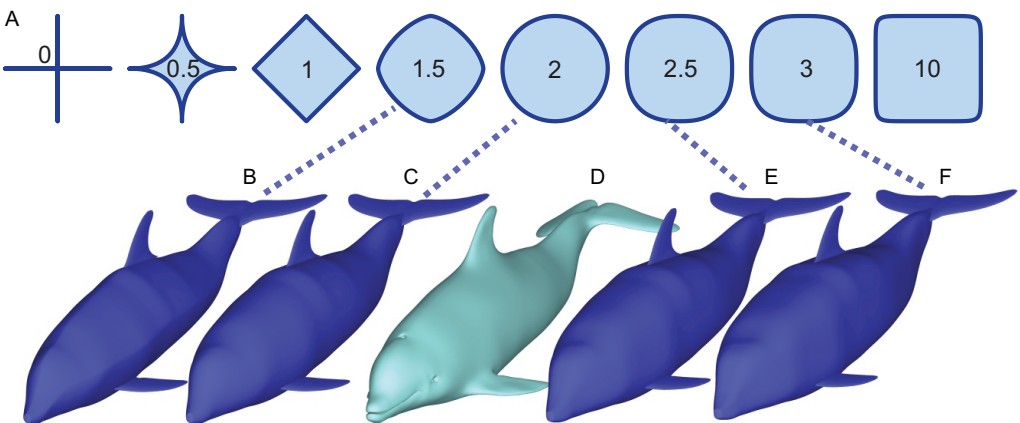

**Figure 1 How superellipses of different exponent values are used to bracket the true volume of a marine vertebrate.** (A) Variations of superelliptical shapes, with numbers being the exponents used to produce respective shapes. (B) A skinny dolphin model with an n value of 1.5 based on the silhouettes from D. (C) Same with an exponent of 2.0. (D) 3D model of *Tursiops truncatus* (model 61 from digitallife3d.org). (E) A fat model with an exponent of2.5. (F) Same with an exponent of 3.0.

for parts of computation: imager (*Barthelmé & Tschumperlé, 2019*), locfit (*Loader, 1999*), Morpho (*Schlager, 2017*), plot3D (*Soetaert, 2022*), rgl (*Murdoch, 2001*), and Rvcg (*Schlager, 2017*). It is open-source and provided under GNU General Public License v3.0. A repository for the package, including the code and a tutorial, is found at: https://github.com/rmotani/paleomass.

## Aim

Paleomass aims to estimate the body volume of a marine vertebrate with a straight body axis. The volume is converted to a mass by assuming the average body density that can be specified by the user. The body surface area is also estimated simultaneously.

## Principle

Paleomass aims to bracket the true body volume of a marine vertebrate between those of two 3D models (Fig. 1). Each of the two models is not the best mean estimate of the true body shape, but one is expected to have a volume slightly larger than the true body volume (Figs. 1E *vs* 1D), and the other slightly smaller (Fig. 1C). The models are based on the same set of orthogonal body silhouette images and therefore appear identical in completely dorso-ventral or lateral views, but have different cross-sectional shapes and differs in coronal view.

The cross-sectional shape is based on superellipses (Fig. 1A), which are mathematical expansions of ellipses. Whereas ellipses are defined as:

$$(x/a)^2 + (y/b)^2 = 1$$

where x and y are the major and minor axes and a and b are the major and minor radii, respectively, superellipses are defined by an equation:

$$|x/a|^n + |y/b|^n = 1 \tag{1}$$

where $n > 0$. When $n = 2$, a superellipse becomes an ellipse (Fig. 1). As n decreases from 2, the superellipse approaches a diamond shape as n approaches 1 and then a cross shape as it approaches 0. If n increases beyond 2, the superellipse approaches a rectangle.

It is known that a typical body cross-section of a vertebrate can be approximated by a superellipse or a combination of two halves of different superellipses (*Motani, 2001*; *Snively et al., 2019*). The true body cross-sections of marine tetrapods are usually found to be bracketed by two superellipses, one with $n = 2$ and the other with $n = 3$ (but see Validation below for a narrower range). For fish, the two exponents are $n = 1.5$ and $2.5$. Therefore, the true volume of a marine vertebrate can be bracketed by making two 3D models with these two boundary superelliptical shapes, depending on the clade (Fig. 1).

## Overall workflow

Paleomass first reads in the data from raster images and command line options, based on which it computes 3D mesh models for the main body and each of the fins and flippers separately. Two mesh models are made for the main body, with different superelliptical exponents of choice. The volume and surface area of each mesh model are computed and summed to give two total estimates, with different main body models. Optionally, these meshes are assembled to make a complete 3D mesh model. The assembled models and each part model can be saved as 3D polygon meshes, respectively.

## Coordinate system

Modeling and computation take place in a three-dimensional Euclidean coordinate system. The x axis is set as the bilateral axis with the right side of the body being the positive side. The y axis is the dorsoventral axis with the dorsal direction being positive, while the z axis is the antero-posterior axis, which may also be called the body axis hereafter, with the tip of the snout being the origin and the posterior direction being positive.

## User supplied data

The users need to supply the shape and size of the animal to be modeled. First, the shape is supplied as a set of silhouette raster images, such as JPEG or PNG, one for each fin/flipper/cephalofoil and a pair for the body (*e.g.*, Fig. 2C)—cephalofoil refers to the "hammer" structure of the hammerhead sharks. These images need to have the same pixel size, *e.g.*, if each side of pixel is 0.001 m in one image, then this pixel side length should be the same in all other images. It is recommended to have at least 3,000 pixels along the body axis of these images (see Validation below), rather than 1,000 as originally suggested (*Motani, 2001*). The body images are in lateral and ventral views, respectively, with all fins, flippers and cephalofoils removed. For each fin, flipper, and cephalofoil, a planar view is required. Second, the length of the body axis as represented in the body images after the removal of the fin/flipper/cephalofoil is supplied through a command line option, in meters.

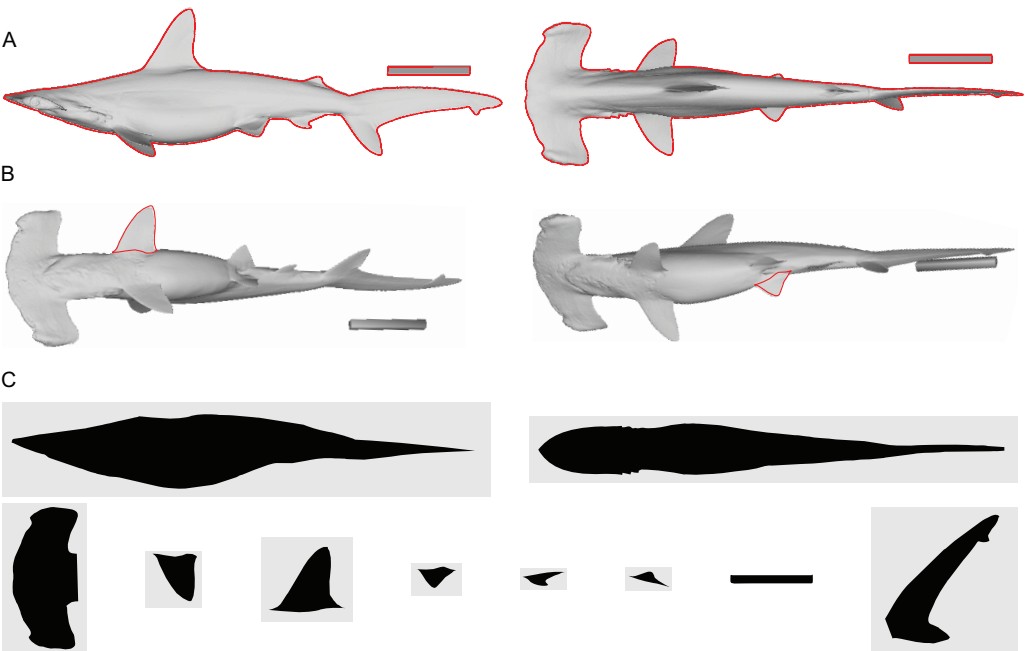

**Figure 2 Shape input images from *Sphyrna lewini*.** (A) Orthogonal views of the target animal, with the overall outlines traced in red. (B) Planar views of fins that are angled in A, with fins in question outlined in red. (C) Input images for Paleomass based on A and B, where fins are separated from the main body. Scale bar in 10 cm. A resulting Paleomass mode is found in Fig. 3E. Image source: A and B are orthographic projections of a 3D model from FFish.asia (*Kano et al., 2013*; https://sketchfab.com/3d-models/scalloped-hammerhead-shark-s-lewini-5de0eec2e8e0462f9a856124761e0ed8; CC BY 4.0, https://creativecommons.org/licenses/by/4.0/).

Paleomass accepts the following types of fins and flippers: pectoral fin/flipper, pelvic fin/flipper, caudal fin, dorsal fin, second dorsal fin, and anal fin. Not all fins/flippers have to be present. This versatility allows for different body architectures to be modelled (Fig. 3).

## Computation steps for main body

The computation of a 3D model and its volume for the main body follows the steps below.

(1) The lateral and dorso-ventral silhouettes of the main body of the animal in question are read from raster image files (Figs. 4A and 4B).

(2) The outlines of these silhouettes are digitized as coordinates (Figs. 4C and 4D), which are then optionally smoothed through interpolation with local regression using the locfit() function (*Loader, 1999*). By default, a nearest neighbor parameter of 0.1 and a constant component of 0 is used for local regression but the former value is user adjustable. The smoothing allows coordinates to take non-integer values and therefore prevents step-like appearance of the final 3D model (Figs. 4G and 4H) that often gives rise to non-manifold edges and triangles that cause errors later on.

(3) The transverse and dorsoventral diameters of the main body are calculated from the coordinates for each pixel position along the body axis (Figs. 4C and 4D). There are much less than 3,000 lines in Figs. 4C and 4D for visualization purposes but the actual calculations are done for each pixel point along the body axis, *i.e.*, there would be 3,000

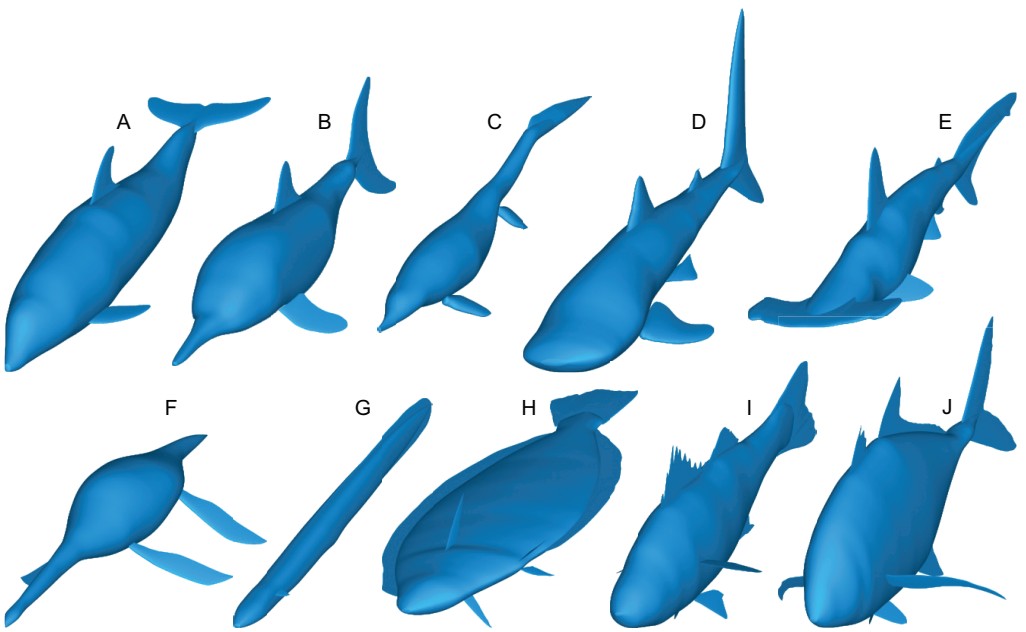

**Figure 3 Range of body morphologies modelled by Paleomass.** (A) *Tursiops truncatus.* (B) *Stenopterygius quadriscissus.* (C) *Chaohusaurus chaoxianensis.* (D) *Rhincodon typus.* (E) *Sphyrna lewini.* (F) *Plesiopterys guilelmiimperatoris.* (G) *Anguilla marmorata.* (H) *Eopsetta grigorjewi.* (I) *Latolabrax japonicus.* (J) *Caranx sexfasciatus.*               

pairs of transverse and dorsoventral diameters in the input body images have 3,000 pixels along the body axis.

(4) Based on these coordinates and diameters, a superellipse is drawn per segment (Figs. 4E and 4F), *i.e.*, body mages with 3,000 pixels along the body axis will result in 3,000 superellipses. Each superellipse has 181 vertices around its perimeter so that there is one vertex per every 2° of angular displacement around the center, with the first and last vertices overlapping—these two vertices will be merged later to make the model watertight, reducing the number of vertices per segment to 180. The number of vertices per segment is user adjustable. The exponent for the superellipse (n in Eq. 1) is also set by the user, *e.g.*, 2 for one model (*e.g.*, Fig. 4E) and 2.4 for the other (Fig. 4F) for marine tetrapods.

(5) A tip is added at each of the anterior and posterior ends of the body to help make the model watertight at a later stage. These tips are small superelliptical disks with a tiny radius of $10^{-4}$ pixels. They do not affect the computation of volume and surface area. The radius of the tip is user adjustable.

(6) Superellipses from steps 4 and 5 are connected as a 3D mesh (Figs. 4G–4J).

(7) Small holes at the tips of the body are closed by merging closely located vertices within distances of $10^{-4}$ pixels or less, and then the whole mesh is cleaned for duplicate faces and non-manifold faces and vertices by vcgClean() function (*Schlager, 2017*). Cleaning may fail if smoothing is skipped at step two, leaving non-manifold edges that would prevent accurate volume calculation. Also, sticky non-manifold edges may result from having low-resolution input images—having 3,000 rather than 1,000 pixels are necessary along the body axis would help prevent this unintended error.

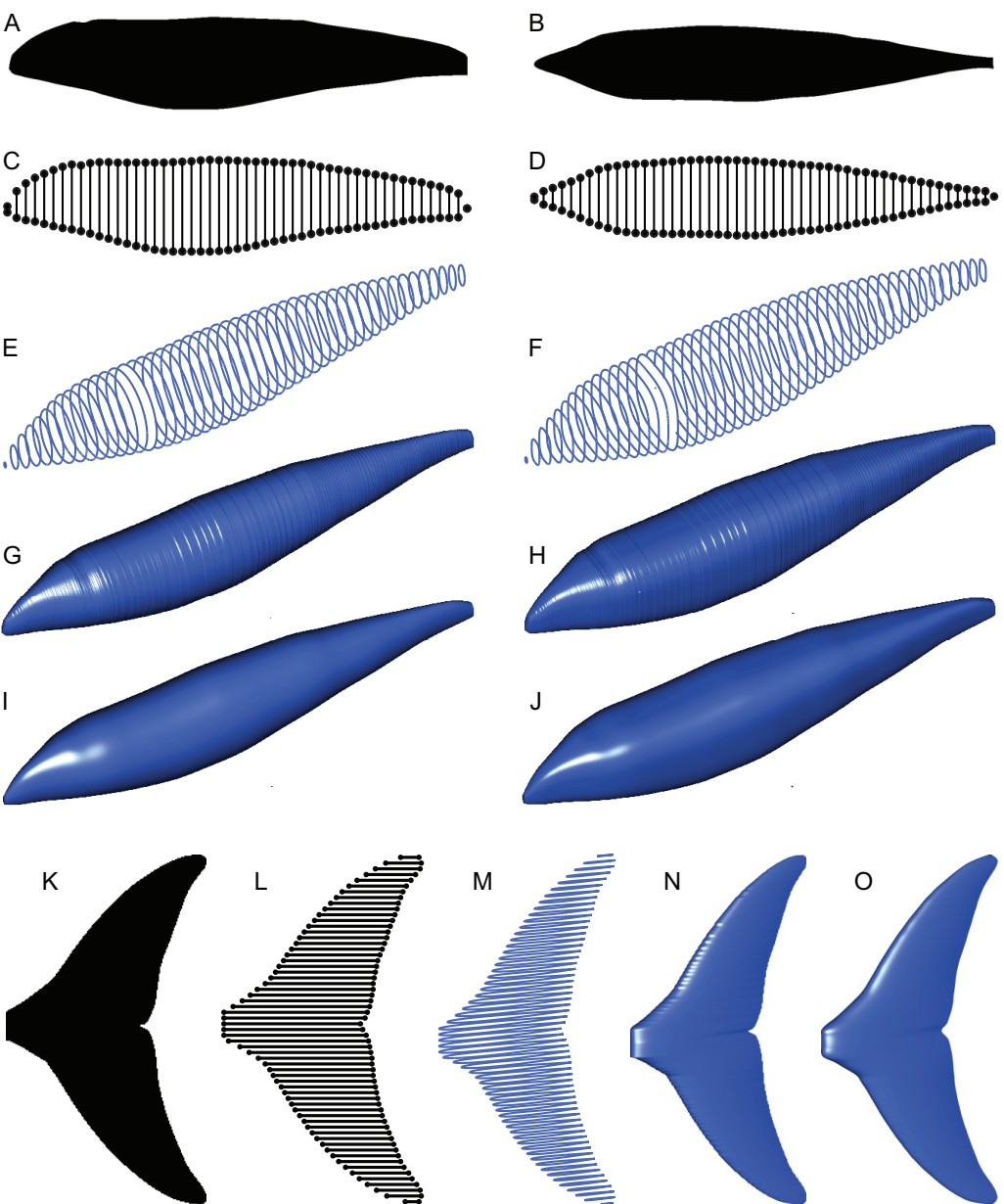

**Figure 4 Computation process of main body and fin/flipper 3D meshes with examples from *Cephalorhynchus heavisidii*.** (A) Lateral silhouette image input. (B) Dorso-ventral silhouette image input. (C) Coordinates around A in dots, with dorso-ventral diameters in lines, down-sampled to one in every ten coordinates for visualization purposes. (D) Same as C but based on B. (E) Serial superelliptical sections based on diameters from C and D, with an exponent of 2, downsampled at the same rate as in C. (F) Same as E but with an exponent of 3. (G) 3D mesh combining all superelliptical slices as in E but without downsampling. (H) Same as G but based on F. (I) Same as G but with interpolation with local regression with a nearest neighbor parameter of 0.1. (J) Same as H but with interpolation with local regression. (K) Planar silhouette image input. (L) Coordinates around A in dots, with chords in lines. Downsampled to one in every five slices for visualization purposes. (M) Serial foil section based on NACA 0020, downsampled at the same rate as in L. (N) 3D mesh that connected serial foil sections as in C but without downsampling. (O) Same as C but with interpolation with a nearest neighbor parameter of 0.05. (M)–(O) are slightly tilted for visualization purposes and thus appear narrower anteroposteriorly than K–L.

(8) The volume and surface area of the model are measured by vcgVolume() and vcgArea(), respectively (*Schlager, 2017*). These functions use the open source vcglib (https://github.com/cnr-isti-vclab/vcglib), which cites *Mirtich (1996)* for volume calculation algorithm. Initially, Paleomass calculates the volume and surface in cubic pixels and square pixels, respectively, where pixel size is as in the input images. These values are then converted to $m^3$ and $m^2$s using the body axis length provided by the user, in combination with the number of pixels along the body axis in the input images.

## Computation steps for fins and flippers

The computation of a 3D model and its volume for a fin or flipper follows the steps below.

(1) The planar image of a fin is read from a raster image (Fig. 4K).

(2) The outline of the image is digitized as coordinates (Fig. 4L), and smoothing through local regression is applied as in the main body outline. The default nearest neighbor parameter for local regression is 0.1 but Fig. 4O was produced with a value of 0.05.

(3) NACA 4-digit foil section is drawn at each pixel point along the span of the fin (Fig. 4M). Symmetrical sections without a camber are used. The equation for such a section is given by:

$$y = 5t[0.2969x^{0.5} - 0.126x - 0.3516x^2 + 0.2843x^3 - 0.1015x^4]$$

where x is the position along the chord given as a fraction between 0 and 1, and t is the thickness of the foil relative to the chord in percentages (*Ladson & Brooks, 1975*). The base value of t is set at 10 for the anal and second dorsal fins and 20 for the rest—these values are user adjustable.

When using the base thickness to construct a fin, the thickness distribution along the span becomes proportional to the chord length distribution and thus results in a strange shape. Most importantly, the part of the fin that is supposed to be thickest along the fin span, *e.g.*, proximal end of the pectoral fin/flipper, is not always reconstructed with the maximum thickness. To avoid this, the base thickness is scaled by a thickness envelope calculated with the following steps. First, the point along the span where the maximum thickness is expected is specified as a fraction between 0 and 1. For example, this point would be 0 for pectoral fin/flipper and 0.5 for a symmetric caudal fin. Second, the axis from the thickest point to an end of the fin span is given new coordinates of 1 to 0, with 1 at the thickest point 0 at the distal tip. Lastly, the square roots of these values are calculated to form the thickness envelope to scale the raw thickness based on the chord lengths. For example, at the midpoint between the thickest point and a fin tip, the raw thickness is multiplied by $0.5^{0.5}$ to give a scaled thickness. This scaling was not present in the original Paleomass.

(4) Foil sections from the previous step are connected to produce a 3D mesh (Figs. 4N and 4O), which are then cleaned as in the body mesh.

(5) The volume and surface areas are measured as in the body mesh.

(6) The processes above are repeated for all fins/flippers.
## Computation of cephalofoil

A simple cephalofoil model is implemented to accommodate hammerhead sharks. A cephalofoil mesh is built in the same manner as fins and flippers, in that a series of NACA foil sections as in Fig. 4M are used. However, unlike fins and flippers that gradually thin out toward the tip, the two ends of the cephalofoil, where the eye sockets are located, are thickened.

## Body and fin integration

This process is for visualization purposes only at present and does not affect the volume/area estimation. Paleomass allows adjustment of the position and angle of each fin/flipper relative to the main body through command line options. Specifically, positioning along the x, y, and z axes, as well as rotation around these three axes can be adjusted. Rotations are called pitch, yaw, and roll, around the x, y, and z axes, respectively. Roll is applied first, followed by pitch, and then yaw.

## Mass calculation

Once the volume of each component is estimated, they are summed to give a total volume. If there is overlap among components, then the overlapping part is counted twice. However, such overlap is usually limited compared to the body volume and would not cause a significant error as evident from the validation results given later. It is ideal to find a Boolean union of 3D meshes, which comprises only those surfaces that are visible from outside, but such a function is not yet stably available in R. Future development may allow addition of a Boolean union procedure.

With the total volume estimated, body mass is calculated from the volume by assuming a mean density of the total body. For marine vertebrates with buoyancy control through an air bladder or lungs, it is expected that neutral buoyancy is experienced in at least a part of daily life. The neutral buoyancy near the sea surface would suggest a mean body density of 1.027 $g/cm^3$, and that in pure water is approximately 1 $g/cm^3$ (*Stewart, 2008*). By default, Paleomass uses these two values, although one of them is user adjustable.

The total body density of vertebrates has been controversial—see *Larramendi, Paul & Hsu (2021)* for a recent review, which advocated a value close to 1 for fish, reptiles, and mammals. I make only one point to augment their arguments against unusually small values suggested by previous authors. *Sellers et al. (2012)* used a value of 0.896 $g/cm^3$, which they calculated based on a dataset from a frozen horse reported by *Buchner et al. (1997)*. However, a reexamination of this dataset suggests that the value should be 0.915 $g/cm^3$–0.896 would be derived instead if the limbs from only one side of the body are included in calculation. This last value of 0.915 is almost identical to the density of ice, so freezing of the specimen may have biased the data, *i.e.*, the true value may be close to 1 without freezing.

# VALIDATION

The accuracy of the software was tested in two ways. First, its accuracy under the best condition was tested by geometric objects of known volume and area, a sphere and prolate

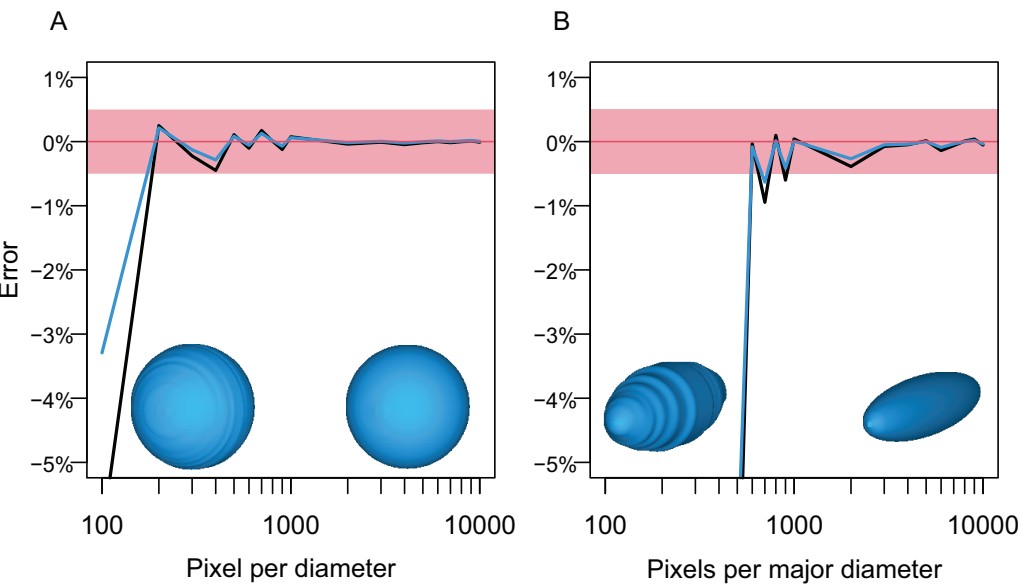

**Figure 5 Errors from volume and surface area estimates for a sphere and prolate spheroid depending on the input image resolution.** (A) Errors from the sphere. (B) Errors from a prolate spheroid whose major axis is five times the minor axis. Blue lines are for the surface area and black for the volume. The independent is the number of pixels along the long axis of the geometry, *i.e.*, pixels per diameter.

spheroid. Second, its ability to bracket the true volume and surface area of actual aquatic vertebrates was tested. In both tests, Paleomass was used with interpolation with local regression enabled.

## Geometric objects

The first test followed the steps below. A circle and an ellipse were drawn in CorelDraw and exported as raster images, respectively, so that the long axis of the object varies from 100 to 10,000 pixels, with an increment of 100 between 100 and 1,000 and 1,000 between 1,000 and 10,000. Then, the volumes of spheres and prolate spheroids based on these images were estimated by using each image as both the lateral and dorso-ventral views for the body in Paleomass, per run. The estimated values were then compared to the true values from parametric equations describing the volume and surface area of spheres and prolate spheroids. The result shows that the error is less than 0.5% in both volume and surface area estimation as long as the resolution of the input image is high, with at least about 800 pixels along the long axis (Fig. 5). However, to stably obtain best results, it is recommended to have 3,000 or more pixels along the body axis (Fig. 5B). Such a high resolution is also beneficial in minimizing unintended production of non-manifold edges as mentioned earlier.

## 3D models of actual animals

The second test is based on 3D mesh models of 25 marine vertebrate species, digitized from actual animals. Only those 3D models that were produced in association with universities (*Kano et al., 2013*; *Irschick et al., 2021*) were used. The species include 20 osteichthyes,

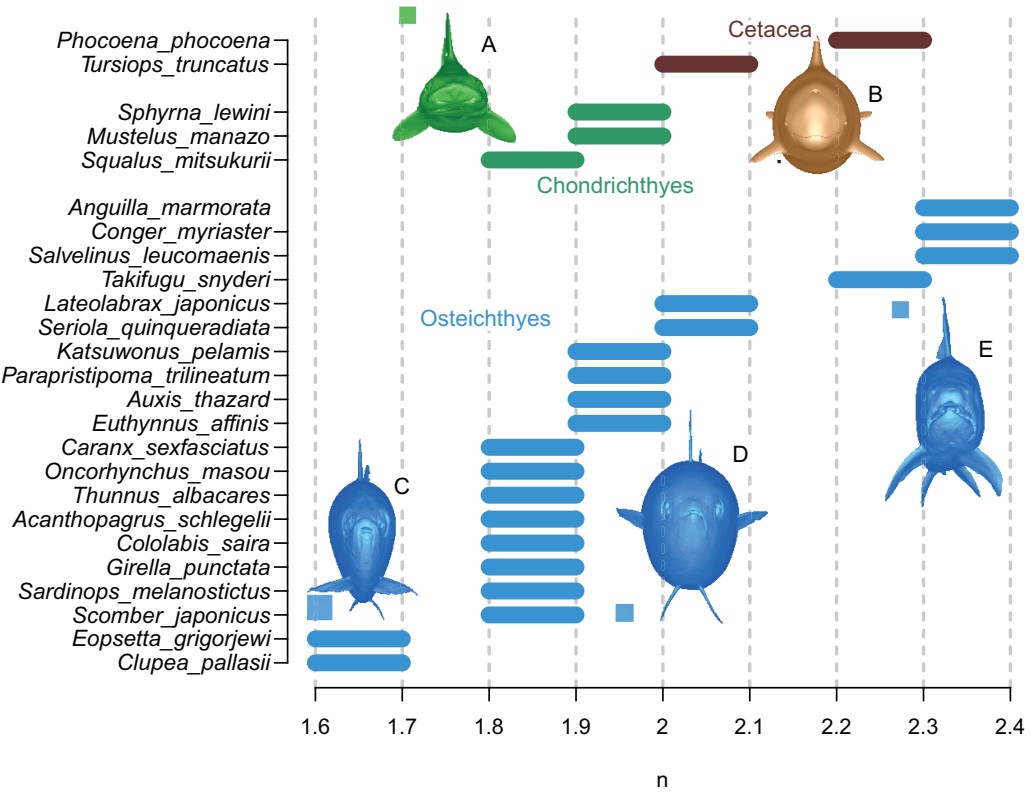

**Figure 6 Optimal superelliptical exponents for 25 species of extant marine vertebrates, with coronal views of five species.** Horizontal bars indicate the range of optimal superelliptical exponents for individual species. Coronal views are given for the following species. (A) *Mustelus manazo*. (B) *Phocoena phocoena*. (C) *Clupea pallasii*. (D) *Auxis thazad*. (E) *Salvelinus leucomaenis*. Species with V-shaped ventral halves of the coronal views, *e.g.*, C, tend to have lower exponent values than those with U-shaped ventral halves, such as E. Squares associated with coronal views are each 1 cm.

three chondrichthyes, and two cetaceans—the list of species used is given in Supplemental Information. The uneven distribution across clades reflects biased data availability that cannot be easily amended.

First, the true volume and surface area of each animal were recorded, after making its 3D model watertight in Meshlab (*Cignoni et al., 2008*). This involved removal of duplicate vertices and faces, followed by an iteration of a sequence comprising deletion of non-manifold edges and self-intersections and filling of the resulting holes. If error-causing borders remained after the iteration, the borders were removed and the iteration sequence was reinitiated. Second, Paleomass estimates of the volume and surface area were calculated based on the lateral and dorso-ventral images of the model, which were captured under orthographic projection in Meshlab (Fig. 2A), together with images from angles that reveal the planar views of individual fins/flippers (Fig. 2B). Attention was paid to not change the magnification between image captures. These images were edited in CorelDraw to separate fins/flippers from the body and then each part was saved as a raster image (Fig. 2C). The image resolution was set so that there are 3,000 pixels along the long axis of

**Table 1 Observed *vs* estimated volume and surface area (SA), together with body axis (BA) length in selected marine vertebrates.**

| Species | BA length | Superellliptical *n* | | Volume (m³) | | | SA (m²) | | |
|---------|-----------|------|------|-----------|----------|---------|-----------|----------|---------|
| | (m) | Low | High | Estimated | Observed | Error % | Estimated | Observed | Error % |
| *Anguilla marmorata* | 0.759 | 2.2 | 2.4 | 7.86E−04 | 7.75E−04 | −1.51% | 9.58E−02 | 9.75E−02 | 1.75% |
| *Conger myriaster* | 0.749 | 2.2 | 2.4 | 7.60E−04 | 7.56E−04 | −0.60% | 9.83E−02 | 1.01E−01 | 2.33% |
| *Salvelinus leucomaenis* | 0.339 | 2.2 | 2.4 | 4.93E−04 | 4.90E−04 | −0.73% | 6.47E−02 | 6.30E−02 | −2.86% |
| *Takifugu snyderi* | 0.127 | 2.2 | 2.4 | 4.85E−05 | 4.85E−05 | −0.06% | 1.12E−02 | 1.08E−02 | −4.03% |
| *Phocoena phocoena* | 1.436 | 2 | 2.3 | 5.15E−02 | 5.09E−02 | −1.14% | 1.18E+00 | 1.04E+00 | −13.24% |
| *Tursiops truncatus* | 2.369 | 2 | 2.3 | 1.63E−01 | 1.67E−01 | 2.31% | 2.47E+00 | 2.42E+00 | −1.77% |
| *Lateolabrax japonicus* | 0.433 | 1.8 | 2.1 | 1.03E−03 | 1.00E−03 | −2.35% | 1.01E−01 | 9.83E−02 | −2.28% |
| *Seriola quinqueradiata* | 0.528 | 1.8 | 2.1 | 1.99E−03 | 1.95E−03 | −1.99% | 1.30E−01 | 1.32E−01 | 1.52% |
| *Katsuwonus pelamis* | 0.437 | 1.8 | 2.1 | 1.72E−03 | 1.72E−03 | 0.32% | 1.08E−01 | 1.09E−01 | 0.60% |
| *Parapristipoma trilineatum* | 0.198 | 1.8 | 2.1 | 1.49E−04 | 1.48E−04 | −0.74% | 2.68E−02 | 2.74E−02 | 2.14% |
| *Auxis thazard* | 0.332 | 1.8 | 2.1 | 6.00E−04 | 5.96E−04 | −0.76% | 5.33E−02 | 5.35E−02 | 0.37% |
| *Euthynnus affinis* | 0.351 | 1.8 | 2.1 | 7.31E−04 | 7.25E−04 | −0.89% | 6.66E−02 | 6.63E−02 | −0.55% |
| *Caranx sexfasciatus* | 0.228 | 1.8 | 2.1 | 2.27E−04 | 2.34E−04 | 2.84% | 3.58E−02 | 3.42E−02 | −4.54% |
| *Oncorhynchus masou* | 0.422 | 1.8 | 2.1 | 9.32E−04 | 9.58E−04 | 2.70% | 9.07E−02 | 9.18E−02 | 1.16% |
| *Thunnus albacares* | 0.395 | 1.8 | 2.1 | 1.03E−03 | 1.06E−03 | 2.66% | 8.47E−02 | 8.40E−02 | −0.85% |
| *Acanthopagrus schlegelii* | 0.209 | 1.8 | 2.1 | 2.41E−04 | 2.47E−04 | 2.26% | 3.99E−02 | 3.98E−02 | −0.32% |
| *Cololabis saira* | 0.318 | 1.8 | 2.1 | 1.31E−04 | 1.34E−04 | 1.70% | 2.64E−02 | 2.65E−02 | 0.46% |
| *Girella punctata* | 0.220 | 1.8 | 2.1 | 3.18E−04 | 3.22E−04 | 1.32% | 4.64E−02 | 5.02E−02 | 7.49% |
| *Sardinops melanostictus* | 0.177 | 1.8 | 2.1 | 7.64E−05 | 7.70E−05 | 0.73% | 1.49E−02 | 1.48E−02 | −0.51% |
| *Scomber japonicus* | 0.280 | 1.8 | 2.1 | 2.36E−04 | 2.39E−04 | 1.18% | 3.24E−02 | 3.29E−02 | 1.35% |
| *Eopsetta grigorjewi* | 0.275 | 1.6 | 1.7 | 2.80E−04 | 2.81E−04 | 0.07% | 6.06E−02 | 5.95E−02 | −1.96% |
| *Clupea pallasii* | 0.221 | 1.6 | 1.7 | 1.33E−04 | 1.31E−04 | −1.34% | 2.37E−02 | 2.42E−02 | 2.22% |
| *Sphyrna lewini* | 0.565 | 1.8 | 2 | 9.41E−04 | 9.39E−04 | −0.24% | 1.14E−01 | 1.26E−01 | 8.96% |
| *Mustelus manazo* | 0.380 | 1.8 | 2 | 1.96E−04 | 1.94E−04 | −1.11% | 3.72E−02 | 3.81E−02 | 2.25% |
| *Squalus mitsukurii* | 0.654 | 1.8 | 2 | 1.74E−03 | 1.76E−03 | 0.80% | 1.46E−01 | 1.46E−01 | 0.53% |

the main body. Paleomass estimates were made for superelliptical exponents (n in Eq. 1) from 1.5 to 3.0 by an increment of 0.1. Finally, the Paleomass estimates were compared to the true values to test if the latter were bracketed by any pair of the Paleomass estimates.

The results are summarized in Fig. 6 and Table 1. In all cases, the true volumes of the marine vertebrates were found to be bracketed between Paleomass estimates with superelliptical exponent values of 1.6 and 2.4. Within this range, cohorts are recognized based on how round the ventral half of the body transverse sections are—some species have rounded ventral halves that appear U-shaped (*e.g.*, Fig. 6E) and found toward the right side of the plot, whereas others have sharper ventral halves appearing closer to a V-shape (*e.g.*, Fig. 6C) and located toward the left side. In sharks, this is upside down, *i.e.*, it is the shape of the dorsal halves that may be rounded or Λ-shaped (Fig. 6A). Most osteichthyes in the data have intermediate ventral halves between V- and U-shape (*e.g.*, Fig. 6D) and consequently found in a moderate exponent range of 1.8 to 2.1. However,

unusual forms are found outside of this typical range—those with flattened ventral sides, such as pufferfish and eels, are in the range of 2.2 to 2.4, whereas those with exceptionally compressed cross-sections with V-shaped ventral halves, such as flatfish and a small herring, are in the range of 1.6 to 1.7. Cetaceans, with their transverse sections rounded ventrally, have a high optimal exponent range of 2.0 to 2.3. Sharks tend to have Λ-shaped dorsal halves but this is partly compensated for by the flat ventral halves, resulting in a moderate exponent range of 1.8 to 2.0. The optimal exponent range for the surface area was between 1.6 and 2, when excluding unusual forms such as flatfish and pufferfish. These optimal ranges mostly overlap the previously suggested ranges (*Motani, 2001*) while being narrower and better defined.

The accuracy of Paleomass estimates was computed in the following manner. Paleomass provides a range of estimates rather than a single mean estimate, while the latter would be required to compute accuracy. Therefore, the mean of the upper and lower bounds of the estimated volume range was used as a single estimate of the volume to facilitate error calculation. With this treatment, the mean and maximum absolute estimation errors are 1.33 and 3.15% across 25 species, using the cohort-specific superelliptical exponent ranges of 2.0–2.3 for cetaceans, 1.8–2.0 for sharks, 1.8–2.1 for typical fish, 2.2–2.4 for U-shaped fish, and 1.6–1.7 for V-shaped fish. When applying the more inclusive exponent range of 1.6–2.4 to all species, the errors increase to 4.61 and 7.21%, respectively. For the surface area, the mean and maximum absolute error are 2.64 and 10.5%, respectively, when using the same inclusive range of 1.6–2.4.

## DISCUSSION

The validation results suggest that Paleomass successfully brackets the true volume and surface area of marine vertebrates when the body silhouettes are known—the software has high accuracy, with a mean absolute error of 1.33%. At the same time, there are limitations to the software package. Paleomass is designed for marine vertebrates with straight body axis and cannot manage lateral concavities in body shape or dorso-ventral concavities in fins/flippers. Also, as stated earlier, the software lacks the capability for Boolean union of body part meshes until such becomes stably available in R. Finally, the accuracy of body mass estimates depends on that of the body outline images, as well as the choice of superelliptical exponent and mean body density.

The accuracy of body outline information merits a discussion. There is a paucity of body outline information in the fossil record in general: some fossil species, such as the ichthyosaur *Stenopterygius* and *Aegirosaurus* (*Motani, 2005*; *Delsett et al., 2022*), are occasionally preserved with body outlines but the majority of species lack such information. For species without body outlines preserved, outlines are often drawn around the skeleton, usually without strict accuracy control. Therefore, the accuracy of body mass estimation for those fossil vertebrates would be lower than that of the software itself because of additional errors introduced while body outlines are reconstructed around the skeleton. One way to remedy this problem may be to employ the minimum hull approach

of *Sellers et al. (2012)*, where the body mass is estimated by multiplying the minimum skeletal hull volume by an empirical ratio between such volumes and the actual volumes in extant mammals. In the present case, the volume of an animal may be estimated from a set of orthogonal minimum skeletal hull silhouettes, provided that the ratio between Paleomass estimates from such silhouettes and the true volume is known. However, derivation of such a ratio would require a broad taxonomic sample of CT scan data that records both the skeletons and body surface of individuals. At present, most publicly available CT scans of marine vertebrates are based on liquid preserved and sometimes eviscerated individuals that do not retain the original body outlines (*e.g.*, MorphoSource. org, *Kamminga et al., 2017*), making it difficult to obtain sufficient data. This possibility may be pursued in the future as more data are added to public data repositories.

At present, it is difficult to assess how much loss of accuracy would result from body outline reconstruction errors based on fossils. However, even if the error level increases by ten- to twenty-fold compared to that from the Paleomass software alone, the total error level would still be comparable to those of other body mass estimation methods for fossil vertebrates. For example, the minimum hull approach had 11–20% errors when applied to primates (*Brassey & Sellers, 2014*), whereas the mean absolute error is 26.35% in the regression-based body mass estimation of terrestrial vertebrates, with the maximum absolute error being about 300% based on the data in *Campione & Evans (2012)*.

Paleomass fills the niche left by other body mass estimation methods. It is applicable to animals for which limb-based regression methods are not suitable, as noted earlier for marine vertebrates. It also enables body mass estimation from flattened fossils, which would supply the body outline images but not a 3D skeletal model necessary for minimum hull construction—again, marine vertebrate fossils tend to be flattened. Application to flattened fossils would depend on the availability of two conspecific individuals with almost identical sizes, exposing the body from two different angles, as in *Stenopterygius* reconstructed by *Motani (2001)*. Overall, Paleomass is a viable alternative to existing body mass estimation methods for fossil vertebrates.

## CONCLUSIONS

Paleomass allows estimation of body volume and surface areas of marine vertebrates with straight body axis through bracketing with 3D models with superelliptical cross-sections. The 3D models are built based on orthogonal silhouettes of the animal in question, which are supplied by the user as raster images. The volumes are converted to body mass by assuming a total body density, which may be the seawater density (1.027 g/cm$^3$) for forms that use the lungs or air bladders to control buoyancy. Optimal superelliptical values for bracketing are 2.0 and 2.4 for cetaceans, 1.8 and 2.0 for sharks, and 1.8 and 2.1 for most bony fish, although the values may be higher or lower for unusual forms, such as pufferfish and flatfish. When using proper exponent ranges, the errors in volume estimation are about 1.33% on average. The software is open access under GNU General Public License v3.0. at https://github.com/rmotani/paleomass.

## ACKNOWLEDGEMENTS

I thank the following individuals for assessing the Paleomass package prior to public release: Benjamin Faulkner, Kiersten Formoso, Yu Qiao, Nicholas Thurber, and Kazuko Yoshizawa.

### Funding

The authors received no funding for this work.

### Competing Interests

The author declares that they have no competing interests.

### Author Contributions

- Ryosuke Motani conceived and designed the experiments, performed the experiments, analyzed the data, prepared figures and/or tables, authored or reviewed drafts of the article, wrote the software code, built the R package, and made it available through Github, and approved the final draft.

### Data Availability

The data and code are available at the Paleomass repository at GitHub: https://github.com/rmotani/paleomass.

### Supplemental Information

Supplemental information for this article can be found online at http://dx.doi.org/10.7717/peerj.15957#supplemental-information.

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
