# Peer review of "Paleomass for R—bracketing body volume of marine vertebrates with 3D models"

_PeerJ, doi:10.7717/peerj.15957_

## Round 0.1 · original submission · Minor Revisions

Dear author,

Based on the opinions of the three reviewers, I have accepted their decision of 'minor revisions'. All three have some comments that would enhance the final version of the manuscript. If the additions would move the text into being an 'extra long manuscript' then they can be included in online supplementary material.

I look forward to receiving your revised manuscript.

·

Basic reporting

The text is clearly written. I have only made a few suggested wording changes in the text (see attached PDF). Background context is fine. The literature citations are fine.

Experimental design

The Paleomass method is well established, and this MS presents an updated version. The methods are all reasonable and transparent.

Validity of the findings

The author measures the accuracy of his method when applied to two standard shapes - a sphere and a prolate ellipsoid - in figure 5. It would be nice to see images of these test shapes to know what was being tested.

The author reports that the methods outlined in the MS were able to bracket the true volume and surface area of the test aquatic vertebrates (lines 253-254). However, no evidence to support this claim is presented. A table listing the taxa presented in Figure 4 along with their observed surface areas and volumes AND the Paleomass estimates of the areas and volumes needs to be included.

Additional comments

I found this to be an easy and pleasant MS to read. The method will be of use to researchers who need body mass and surface area estimates of animals, but do not have the mathematical/computing skills to generate and and analyze their own digital models. I have no substantial criticisms - just the points mentioned in Section 3 that need to be addressed.

Reviewer 2 ·

Basic reporting

This paper provides a new version of the software Paleomass, updated to fit the R environment and estimate accurate body masses from 3D models produced from silhouettes. It focuses on marine vertebrates and uses the produced 3D reconstructions to calculate mass estimates with a very low error rate. This approach has clear implications for fossil taxa where body reconstructions based on skeletal material often rely on the volume of a produced 3D model

Experimental design

The aim of this research is well and clearly defined.
The ReadMe file in the supplied github link provides a useful tutorial for the package that any reader or future researcher should be able to follow to replicate the research and apply the code to their own modelling work. Furthermore, the details for the package’s computational steps, including tests validating the software’s accuracy, are described in sufficient detail in the main text.

Validity of the findings

As R code is provided, and the results are assessed using two different validation tests that find strong model accuracy in the package. The conclusion is well stated and clearly linked to the original aim of the research.

Additional comments

In general, we enjoyed reading this paper and consider this software will be particularly useful for paleontologists. Below, we include a few minor comments to be addressed or clarified.
- It could be useful to provide some definitions of the most technical terms like serial NACA, cephalofoil and Boolean union, for example in a short table.
- Lines 101-103: n is well defined in the ellipse equations; though it would be good to clearly state what x and y represent in these equations
- Line 141: Figure 4 is cited before Figure 3.
- Line 246-247: The literature cited for body density of sharks and rats is relevant, though are old references. We recommend checking and citing more recent literature to ensure such estimates are up to date. For example, for sharks, which notably use their livers to control their buoyancy, there is a recent paper that generally agrees with Baldridge and found that pelagic sharks have a mean body density of 1.06 g/cm3 (Gleiss et al. 2017).
Reference:
Gleiss, A.C., Potvin, J. and Goldbogen, J.A., 2017. Physical trade-offs shape the evolution of buoyancy control in sharks. Proceedings of the Royal Society B: Biological Sciences, 284, 20171345.

·

Basic reporting

The manuscript is direct, concise, and clear, approaching McNeill Alexander levels Citing several recent reviews of mass estimate methods and body densities will bring the context up to date. Full citations for these papers are given in the review copy. I suggest a few grammatical rearrangements.

Experimental design

The manuscript presents a welcome, updated method for estimating geometric properties of marine vertebrates, in the widely usable format of an R. package. It fills a methodological need, or lapse, since the language behind the original Paleomass program was superannuated.
The methods as described will enable others to apply them to their own research, I was looking forward to explanations of the math used to calculate surface area and volume, and was a bit disappointed that the included R/VCG library functions were not explained. A summary of the mathematical approach would be appropriate.

Validity of the findings

The manuscript includes an exciting validation of the method, which shows convergence on stable estimates when the length of source images reaches 3000 pixels. The honest bracketing approach is more valuable (and realistic) than attempts at point estimates; similar to regression-based superellipe correction factors I've used.

Additional comments

The manuscript is thoroughly lucid and self-contained, and I have just a few suggestions for improvement on the review copy. The manuscript understates the influence of Motani's (2001) introduction of super elliptical cross-sections, which have now been applied to mass property estimates in dinosaurs, including mass moments of inertia. i suggest the most pertinent reference, among several. Volume primitives in Figure 5 will harmonize with the pleasing volume renders of taxa in the other figures.

---

## Round 0.2 · accepted · Accept

Dear Author,

Thank you for your revised manuscript. Based on your response to reviewers .doc I am happy to accept your manuscript for publication.

Shortly, the production staff will contact you to take you through the proofing stages.

Congratulations, and thank you for choosing PeerJ as your publication venue. I hope you will use us again in the future.